# Design of Novel Oligomeric Mixed Ligand Complexes: Preparation, Biological Applications and the First Example of Their Nanosized Scale

**DOI:** 10.3390/ijms20030743

**Published:** 2019-02-10

**Authors:** Rawda M. Okasha, Najla E. AL-Shaikh, Faizah S. Aljohani, Arshi Naqvi, Eman H. Ismail

**Affiliations:** 1Department of Chemistry, Taibah University, Al-Madinah Al-Munawarah 30002, Saudi Arabia; najla1407@hotmail.com (N.E.A.-S.); m.sfm@hotmail.com (F.S.A.); arshi_84@yahoo.com (A.N.); 2Chemistry Department, Faculty of Science, Ain Shams University, Cairo 11566, Egypt; eman_hamed@sci.asu.edu.eg

**Keywords:** ternary complexes, oligomers, nanoparticles, in silico predictions, antimicrobial activity, antitumor behavior

## Abstract

A successful oligomerization of ternary metal complexes, cobalt (II), nickel (II), copper (II), zinc (II), chromium (III) and ferric sulfate (III) with nitrilotriacetic acid (NTA) as a primary ligand and glutamic acid as a secondary ligand, has been demonstrated. The formation of oligomers arose from the presence of the sulfate moiety, which operates as a bridged bidentate ligand that coordinates with other metal moieties. The novel oligomers exhibited octahedral structures, which bonded together through the sulfate moiety. In silico predictions were conducted to gauge the bioactivity, physico-chemical and pharmacokinetic properties. The biological activities of these oligomers as well as their tumor inhibitory behavior have been explored. This work also presents a facile and novel method of preparing these materials in nanosize, using Cetyltrimethylammonium bromide (CTAB) and polyvinyl alcohol (PVA) as capping ligands. The size and shape of the nanomaterials have been confirmed using the transmission electron microscope (TEM) and the scanning electron microscope (SEM).

## 1. Introduction

Coordination compounds and macromolecules are an eminent class of material that are encountered daily [1,2,3,4,5,6,7]. The sensible selection of the metal connectors and the bridging ligands permits the utilization of these materials in areas, such as ion exchange, adsorption and separation processes [8,9,10,11,12]; heterogeneous and biomimetic catalysis [12,13,14,15,16]; sensor technologies [17,18]; biomedical applications [19,20]; luminescence [17,21,22]; drug delivery [21,23]; and proton conductivity [12].

The metal-organic coordination macromolecules exhibit remarkable stability in comparison to their discrete complexes due to their macrocycle or chelate effect. The synthesis of these coordination oligomers and polymers was accomplished via several successful strategies, such as the metalloligand strategy, pillar-layered and supramolecular building blocks and layers. In the meantime, the synthesis of these materials could be achieved via the conventional synthesis and alternative methodologies.

The selection of the ligand precursor has a great effect in tuning the properties of the obtained materials. For instance, the presence of nitrogen and oxygen donor ligands as building blocks in the macromolecules architecture triggered exceptionally high stability [24,25], resulting in their application as probes for biological redox centers to be quite desirable [26]. In addition, the study of the structure of their model ternary complexes provides information on how the biological systems attain their specificity and stability [27]. From this perspective, two complexing agents have been selected to synthesize a new family of coordination oligomers, containing mixed ligand complexes. Nitrilotriacetic acid (NTA) has been used as the primary ligand, which is a tridentate aminocarboxylic acid that bears three identical carboxylic groups. Interestingly, coordination can also occur via the tertiary nitrogen atom in order to complete the metal polyhedron. Therefore, NTA is known for its strong complexes with several divalent and trivalent metal cations. This coordination behavior alters metals’ bioavailability, mobility and persistence in the environment [28].

Furthermore, NTA has also merged into the complexes’ skeletons for several purposes such as biotechnology, especially in the protein purification technique (IMAC) [29,30]. It has also been employed in other applications, such as food, metal finishing, pharmaceutical, cosmetic, photographic, tanning, metal plating and cleaning operations, textile manufacture, paper and cellulose production [31,32,33]. Moreover, as a result of its strong coordination with divalent and trivalent metal cations, NTA has been widely treated in domestic and industrial domains to remove metal cations and are, subsequently, discarded in wastewaters [34].

The incorporation of amino acids in mixed ligands is also quite intriguing, owing to the fact that they represent important chelating agents, which are involved in various living cycles. Glutamic acid has been selected to be the secondary complexing agent due to its significant biological value. For example, glutamic acid contributes in metabolic reactions more than any other amino acids. It is also deemed as a source of glucose, and it maintains the level of glucose in the blood normally [35]. L-Glutamic plays a vital role in various biochemical processes at molecular levels [36], and it links the metabolism of carbon and nitrogen [37]. Researches have shown that glutamic acid is a focal component in the cell wall of gram-positive bacteria and is also essential for other microorganisms to grow on [35].

Herein, we are presenting the synthesis and characterization of a new series of coordination oligomers, containing mixed ligand complexes with nitrilotriacetic and glutamic acid. The metal precursors perform a vital role in the oligomerization rather than forming discrete complexes. It is notable that these complexes could be prospective candidates for biomedical applications. For this matter, this report also discusses the antimicrobial and antitumor activities of these oligomers. Moreover, the in silico parameters viz. bioactivity, physicochemical and pharmacokinetic property predictions will also be appraised.

In the present biomedical domain, the in silico or computer-aided drug designing (CADD) notion is exploited to assist and speed up hit identification, to elect a hit-to-lead and to screen out vast compound libraries into smaller groups of predicted active compounds. The potential use of any drug aspirant like a therapeutic agent primarily depends on their pharmacodynamic and pharmacokinetic traits. The CADD approach offers various techniques to gain an insight on the bioactivity, physicochemical and pharmacokinetic properties of the targeted drug candidates [38,39].

Recently, a tremendous interest has been devoted to the synthesis of nanoscale materials. This interest has been derived from their distinctive properties, including catalytic, optical, semiconductive, magnetic, treatment of human diseases, drug delivery and others [40,41,42,43]. Consequently, this class of materials ensures great expectations for their applications in different fields of material science and technology.

In literature, great efforts provide access to the nanoscale materials, ranging from inorganic metal clusters to custom-built single molecules [44]. Ternary metal complexes have played a significant role as precursors in the production of nanoparticles, using different techniques, such as thermal decomposition, UV irradiation, sonication, microwave, etc. [44]. Nevertheless, literature has not afforded a clear example of the synthesis of nanoscale ternary complexes though there are a few that have alleged to have done so; however, the authors did not supply clear evidence to confirm the actual size and shape of their molecules. Moreover, the obtained nanoparticles suffered from great agglomeration [45,46,47].

In this work, we introduce a prosperous attempt to synthesize the first example of ternary metal complexes, containing oligomers in nanoscale. The nano-macromolecules were acquired using two different capping agents, polyvinyl alcohol (PVA) and cetyltrimethyl ammonium bromide (CTAB). The size and shape of the obtained nanoparticles have also been verified. 

## 2. Results and Discussion

A new series of ternary metal complexes of cobalt (II), nickel (II), copper (II), zinc (II), chromium (III) and ferric sulfate (III) with nitrilotriacetic acid (NTA) as a primary ligand and glutamic acid (Glu) as a secondary ligand has been synthesized. All the desired complexes were prepared in a slightly acidic medium with a molar ratio (1:1:1) of metal ions, NTA and glutamic, respectively. The new complexes revealed applicable solubility in water and formed slightly weak effervescence with the evolution of carbon dioxide upon the addition of potassium carbonate, which agreed with their measured pH values (Table 1). The molecular masses of all complexes insinuated the presence of the sulfate group (SO_4_^2−^) as a ligand (Table 1). A simple test of the BaCl_2_ solution was performed on all the complexes as evidence that the sulfate groups act as ligands not as counter ions [48,49].

The pH values of all oligomers were measured and found to be in the range of 2.3–4.4. At this pH range, the HNTA^−2^ species coordinated via its three sites (one N and two COO^−^) [48,50]. Meanwhile, the glutamic acid, which converted to azwitter ion form (H_3_N^+^ CH_2_COO^−^), worked as a monodentate ligand and coordinated via its carboxylic oxygen [51]. Thus, the octahedral structure of each monomeric unit is formed via two oxygen and one nitrogen atoms of the HNTA^−2^ moiety combined with the carboxylic groups of the glutamic acid and the two halves of the sulfate groups, which bind to the metal ions as bridged bidentate ligand, forming the oligomeric materials [49]. The proposed structure of this oligomeric series is illustrated in Figure 1. It is also crucial to note that there is a possibility of forming hydrogen bonds through oligomer structures, Figure 1.

### 2.1. Infrared Spectra

The FTIR spectra of all the novel material have been performed in the range of 200–4000 cm^−1^ and confirm the validity of the proposed structures. For instance, the presence of non-coordinated free carboxylic groups of H_3_NTA and Glu of metal complexes was presented clearly at 1724 cm^−1^ in oligomer **4** while the rest of the series exhibited an overlapping of these values with the coordination of COO^−^ due to the formation of the hydrogen bonds (Table 2). The vibrational frequencies of the COO^−^ group displayed a suitable shift because of the complexation with metal ions. In addition, the –NH_3_^+^ bands in the zwitter ionic form appeared in the free glutamic acid at 3062 cm^−1^, which shifted it to a very small value upon complexation (Table 2) [52,53]. The CN vibration band appeared at 1206 and 1242 cm^−1^ for NTA and Glu, respectively [54,55]. Upon complexation, this value shifted to be in the range of 1026–1088 cm^−1^ due to the coordination of a nitrogen atom with the NTA (Table 2). The formation of the new coordinated oligomers has also been established by resolving the M–O and M–N frequencies, which disclosed their vibrational frequencies in the range of 318–395 and 550–557 cm^−1^, respectively. Moreover, the FTIR analysis also illustrated that the SO_4_^−2^ ion forms a bridge bidentate ligand between the complex with a C_2v_ symmetry [56]. This behavior is supported by the appearance of bands at ν_1_ at approximately 981–997 cm^−1^ and ν_2_ at approximately 443–482 cm^−1^ while ν_3_ split into 1026–1088, 1169–1193 and 1102–1153 cm^−1^ and ν_4_ split into 551–598, 604–629 and 641–677 cm^−1^ for the new molecules, which provided a great substantiation for the oligomerization process.

### 2.2. Mass Spectra

The mass spectra of the acquired oligomers were recorded and provided good insight that supported the proposed molecular formula of the monomeric complexes, as well as confirming their architectures, Table 3. For instance, all the oligomers displayed mass peaks (*m*/*z*) at 497, 499, 495, 509, 499 and 425 for Co, Ni, Cu, Zn, Cr and Fe, respectively. These molecular ionic peaks were attributed to the suggested formula weight of the monomeric unit of these oligomers. Moreover, the spectra revealed continuous fragmentation processes after the suggested mass peak indicated that the monomolecular formulas of the prepared ternary metal complexes were repeated. This behavior provides a strong verification of the formation of the oligomeric structure of these materials.

### 2.3. UV-Vis Spectra

The UV-Vis spectra of the prepared oligomers with the ternary metals complexes were recorded in bi-distilled water with the concentration of 10^−2^ M (Figure 2). The electronic spectrum of the Co(II) complex showed an overlapping band at λ_max_ = 511 nm (19,569 cm^−1^ ), which is assigned to the ^4^T_1g_(F) →^4^T_1g_(P) transition and assumed their tetragonal distorted octahedral geometry within the chain [50,57]. Contrastingly, the spectrum of the Ni(II) containing oligomer revealed three bands at 13,531, 15,822 and 25,733 cm^−1^*,* which are assigned to ^3^A_2g_→^3^T_2g_, ^3^A_2g_→^3^T_1g_(F) and ^3^A_2g_→^3^T_1g_(P), respectively, and correspond to the perfect octahedral sites [50,57,58,59].

The Cu(II) complex displayed a broad band at 12,484 cm^−1^*,* which is appointed to the ^2^E_g_→^2^T_2g_ transition with an expected splitting of this state as a result of the tetragonal distortion of the octahedral Cu(II) ion, d^9^ [50,57,59]. Meanwhile, the Zn(II) moieties did not indicate any d–d transitions but displayed charge transfer bands at 50,000 cm^−1^ [57,60].

The Cr(III) containing complex exhibited two bands at 24,691 and 17,857 cm^−1^. These bands may be allocated to the^4^A_2g_(F)→^4^T_1g_(F) and ^4^A_2g_(F)→^4^T_2g_(P) spin, allowing d–d transitions, respectively [57,58]. While, the Fe(III) oligomer exhibited bands at 34,482 and 13,586 cm^−1^*,* which may correspond to the ^6^A_1g_→^4^A_1g_(G), ^4^E_g_(G) and ^6^A_1g_(S)→^4^T_1g_ (G) transition in the tetragonal distorted octahedral geometry within the oligomer chain, respectively [57,58].

### 2.4. Magnetic Measurements

It is prominent that the magnetic moment provides information to characterize the structure of the complexed units within the oligomeric chains (Table 1). For example, the magnetic moment value of the Co(II) complex within the backbone is attained to be 6.49 B.M. This value is higher than the only spin moment for three unpaired electrons (3.89) due to a considerable orbital contribution [50,57,58,59] while the tetrahedral Co(II) complexes have paramagnetic moments of (4.3 ± 0.5 B.M.) [57]. The ternary Ni(II) moieties showed a magnetic moment value of 2.99 B.M., which suggested its octahedral geometry [50,57,58,59].

The Cu(II) mixed-ligand oligomer displayed a magnetic moment value of 1.16. This value corresponds to one unpaired electron, and this offers evidence for the mononuclear structure of the tetragonal distorted octahedral repeating units [50,57,59]. According to the empirical formula of the Zn(II) moieties, which are diamagnetic, a tetragonal distorted octahedral geometry was proposed for this ternary chelate [58]. The magnetic moment value of Cr(III) is found to be 4.96 B.M, indicating its distorted octahedral geometry. Additionally, the Fe(III) moieties revealed a magnetic moment value of 4.16 B.M. with a high spin tetragonal distorted octahedral [57,58,61].

### 2.5. Thermal Analysis

The thermogravimetric analysis (TGA) of the prepared complexes was carried out in a nitrogen atmosphere. The thermal decomposition of all oligomers displayed similar patterns as their ligand precursors. For instance, it is renowned that H_3_NTA is thermally decomposed in two overlapping steps into glycine and maleic acid [49] while glutamic acid decomposes, producing glycine, CO_2_ and C_2_H_4_ [62]. The correlations between the different decomposition steps of the coordinated oligomers with the corresponding weight losses is discussed in terms of the proposed formula of their complexed monomers. Figure 3 illustrates the TGA thermograms of all the new materials.

The weight losses of each chelate are calculated within the corresponding temperature ranges (Table 4). In general, the first weight loss of these oligomers was in the range 25–150 °C, which was accompanied with the evolution of the H_2_S and NH_3_ molecules while the second and main weight loss occurred between 150 and 430 °C and is attributed to the loss of the maleic acid and glycine molecules. The third weight loss occurred within the temperature range over 430 °C, which could be attributed to the releasing of C_2_H_2_, CH_4_, CO_2_ or CO [62]. The end step of the residue in all complexes, usually occurring in above 650 °C except for zinc, corresponded to the thermal decomposition of the sulfate group to the sulphite (SO_3_^2−^) or the hypo-sulphite (SO_2_^2−^) group with the formation of reducing gases such as CO, C_2_H_2_ or C_2_H_4_ in the earlier steps [63].

### 2.6. In Silico Predictions

All the targeted monomeric units of oligomers have been tested for the in silico study in order to evaluate their bioactivity, physicochemical and pharmacokinetic properties. The bioactivity score predictions for six different protein structures viz. the GPCR ligand, ion channel modulator, kinase, nuclear receptor ligand, protease and enzyme inhibition were performed. The bioactivity score profiles of the selected compounds **1–6** is given in Table 5. The values attained indicated the binding affinity of the selected candidates to the mentioned enzymes and receptors (positive values indicate greater affinity while negative values mean low affinity). All the tested compounds have shown positive or good bioactivity towards the selected protein structures, except for the kinase and the nuclear receptor ligand. 

The predicted physicochemical properties of the scrutinized compounds are documented in Table 6. The results disclosed in this table for the specified properties are molecular weight, electronic distribution, hydrogen bonding characteristics, molecular refractivity, water solubility and topological polar surface area (TPSA). All the predicted units are soluble in water with good molar refractivity at approx. 100. The molar refractivity should be between 40 and 130.

The pharmacokinetics of the investigated compounds is revealed in Table 7. The results presented in Table 7 indicated that all of the evaluated compounds manifested a low gastrointestinal absorption (GI), no penetration through blood brain barrier (BBB) and low skin permeation (Log K_p_), and they were not able to inhibit the cytochrome P450 (CYP) involved in metabolism but were predicted to be the substrates for P-glycoprotein (P-gp).

### 2.7. Biological Activity

All the desired oligomers have been tested for the in vitro assay for their antimicrobial activities by the agar diffusion method, using the Mueller–Hinton agar medium for bacteria and the Sabouraud’s agar medium for fungi [64]. The assayed collection included two gram-positive species of pathogenic bacteria: Streptococcus pneumonia and Bacillissubtilis; they also included two gram-negative ones: Pseudomonas aeruginosa and Escherichia coli. The two standard antibiotics, ampicillin and gentamicin, (25 µg/mL) have been used as reference drugs for these assays. The fungi toxicity screening of the ternary complexes was performed against four fungi: Aspergillus fumigatus, Syncephalastrum racemosum, Geotricum candidum and Candida albicans while using Amphotericin B (25 µg/mL) as a reference drug. The mean zone of inhibition in the mm ± standard deviation beyond the well diameter (6 mm) was determined using a 25 µg/mL concentration of the tested complexes. The inhibitory effects of the synthetic complexes against these organisms are given in Table 8.

The in vitro assay revealed that the fungal growth inhibition has varied with the metallic units within the complexes. For instance, it was found that the Cu polymer exhibited significant activity against Aspergillus fumigatus with a Minimum Inhibitory Concentration (MIC) value two times more active than the reference drug. The same oligomer also displayed strong activity against Syncephalastrum racemosum while the Fe complex revealed excellent antifungal activity against Aspergillus fumigatus, compared to the Amphotericin B control. The antifungal activities of the remaining polymers varied from moderate to weak inhibitory effects (Table 8).

The antibacterial activities of the obtained oligomers were also determined in terms of their MIC values (Table 8). For example, the Fe complex showed significant antimicrobial activity against the gram-positive Streptococcus pneumonia bacteria with a MIC of 0.12, which is two times more active than the control, while the Cr and Fe complexes displayed moderate activities against Bacillissubtilis. The inhibitory effects of the remaining coordinated oligomers were very weak and can be negligible. The growths of the gram-negative bacteria Pseudomonas aeruginosa and Escherichia coli were affected by the Cu and Fe complexes, which revealed the MIC values were lower than their control. The Ni oligomers also exhibited strong inhibition against the gram-negative pathogenic bacteria with a MIC value lower than the Gentamicin control. The remaining complexes did not display the same behavior during the assay. 

### 2.8. Anticancer Activity

The cytotoxicity assays of the ternary complexes containing oligomers have been evaluated for their human tumor cell growth inhibitory activities against three cell lines: breast carcinoma (MCF-7), colonic carcinoma (HCT-116) and hepatocellular carcinoma (HepG-2) (Table 9). The obtained results indicated that most of these molecules revealed weak inhibitory against the studied cell lines. This behavior can be observed from the values of the half maximal inhibitory concentration IC_50_, which were discovered to be in the range of 10.7 μg/mL and >50 μg/mL compared to the standard drug Vinblastine (IC_50_ = 2.5, 6.1 and 4.6 μg/mL for HCT-116, MCF-7 and HepG-2, respectively).

### 2.9. Synthesis of Nanosized Coordinated Complexes

The synthesis of the nanosized ternary metal complexes containing oligomers has been achieved using two different capping agents: Cetyltrimethylammonium bromide (CTAB) and polyvinyl alcohol (PVA). The reactions were carried out by mixing equivalent amounts of metal ions, NTA and Glu, followed by adding different concentrations of the capping agents. The obtained precipitates were filtered, washed with ethanol and left to dry in a vacuum desiccator. 

The size of the prepared particles was determined, using a transmission electron microscope (TEM). This methodology was successful in isolating these oligomeric materials as homogenous nanoparticles. The size of the obtained nanoparticles was controlled by changing the concentration of the capping agents. For instance, these complexes were isolated in the 2.17–8.86 nm diameter in the presence of 10^−2^ M CTAB. Figure 4 illustrates that these particles exhibit a homogenous spherical shape, which hangs on a large sphere of the capping materials. In some cases, the TEM images show slanted hexagonal shapes of these particles (Figure 5c), which verify the formation of octahedral structures within the chains. The images displayed a harmonization of some of these nano-complexes with CTAB (10^−2^ M). For instance, copper and zinc containing oligomers exhibited small sizes with homogenous shapes, shown in Figure 4d and Figure 5d, respectively. Upon the dilution of the CTAB concentration from (10^−2^ M) to (10^−3^ M), the nano-complexes displayed larger sizes with less homogeneityas in cobalt, zinc and chromium; however, copper’s particles continually revealed monodispersity (Figure 5a,b). This behavior indicates that a higher concentration of the CTAB within the studied concentrations is considered more suitable for the preparation of these nano-complexes. 

The PVA has also been used as a capping agent for the preparation of the nanosized oligomers with two different concentration values (10^−5^ and 10^−6^ M). Using the (10^−5^ M) PVA, most of the acquired complexes showed modifications in their sizes and shapes. Compared to the case of CTAB, the nano-complexes are found to be larger and less homogeneous. Some TEM images show the agglomeration of the isolated particles with a peduncle shape, such as the chromium complex (Figure 6). This concentration of the PVA allowed for the preparation of nanoparticles with an average size of 3.26–27.06 nm. A diluted concentration of PVA (10^−6^ M) permitted the isolation of smaller nanoparticles. For example, nickel, copper and zinc complexes exhibited nanoparticles with the average diameter of 2.31–16.99 nm (Figure 7). In general, a comparison between CTAB and PVA divulged that both of them have an excellent ability to work as capping agents for this class of nanoparticles depending on their concentrations. 

Some of these complexes have also been viewed using the scanning electron microscope (SEM). Figure 8 illustrates the SEM images of the copper oligomeric material in the presence of (10^−2^ M) of CTAB and (10^−5^ M) of PVA, respectively.

## 3. Materials and Methods

### 3.1. Materials

All chemicals were of the analytical reagent grade (AR) and were used without further purification: CuSO_4_·5H_2_O from Fisher (Loughborough, UK); NiSO_4_·6(H_2_O), Fe_2_(SO_4_)_3_·5H_2_O and ZnSO_4_·7H_2_O from B.D.H (Bristol, UK); Cr_4_(SO_4_)_5_(OH)_2_ and CoSO_4_·7H_2_O from Hopkin & Williams (London, UK); Glutamic acid from s.d. fine Chemical Ltd. (Tamil Nadu, India); Nitrilotriacetic acid (NTA) from Merck (Kenilworth, USA.); Cetyltrimethyl ammonium bromide (CTAB) from Aldrich (Gillingham, UK); and Polyvinyl alcohol (PVA) from Oxford laboratory (Maharashtra, India). The complexes have been prepared in bi-distilled water.

### 3.2. Instrumentation

The microanalyses of carbon, hydrogen, nitrogen and sulfur were determined using the Vario El Elementar. The Ni, Cu, Zn, Co, Cr and Fe percentages were determined by the atomic absorption spectrometry (AAS) using a Perkin Elmer AAs 3100. The IR spectra of the solid complexes were recorded on a Jasco FTIR-300 E Fourier Transform Infrared Spectrometer CsI in the range of 200–4000 cm^−1^.

The mass spectra were carried out to confirm the molecular mass of the obtained complexes using two different instrumentation. The first mass spectra were recorded at 350 °C and 70 eV on the Shimadzu GC/MS-QP5050A spectrometer, which allowed for the calculation of the mass of the repeating units. Further analysis was performed using a triple quadruple mass spectrometry API4500 (ABSciex, Concord, Ontario, Canada) attached to the Exion LC™ chromatographic system (ABSciex, Framingham, MA USA), confirming the presence of oligomer fragmentation. The optimization of the analyte-dependent MS/MS parameters was performed via direct infusion of standards into the MS at a flow rate of 10 μL/min. Data acquisitions were performed with the Analyst 1.6.3 software Hotfixes^®^. The electronic UV-Vis spectra of the metal complexes were recorded in a bi-distilled water solution (10^−2^ M) at room temperature with typical ranges from 190 to 800 nm on Carry 100.

The thermogravimetric analysis (TGA) of the complexes was carried out under a nitrogen atmosphere at a heating rate of 10 °C min^−1^ using the TA instrument model SDT600. The magnetic susceptibilities of the paramagnetic metal moieties were measured at room temperature by the Gouy method using a magnetic susceptibility balance Jhon-son Matthy, Alfa products model No. MKI. The diamagnetic corrections were calculated from the PASCAL’s constants, and Hg(Co(SCN)_4_) was used as a standard. 

The shape and size of the nano-complexes were revealed by the transmission electron microscopy, type JEOL JEM 1400 electron microscope. The TEM images were attained by preparing aqueous solutions of the nano-complexes with a concentration of (10^−4^–10^−6^ M), followed by the deposition onto a carbon-coated copper grid and allowed to dry before the measurement of the observation. 

The Scanning Electron Microscope (SEM) measurements have been performed using the SEM Model Philips XL 30 (FEI company, Hillsboro, OR, USA) with an accelerating voltage of 30 K.V., magnification 10× up to 4000× and a resolution of W.

### 3.3. Synthesis of Ternary Metal Complexes

#### 3.3.1. Synthesis of Divalent Metal Complexes of Co, Ni, Cu or Zn

The divalent ternary metal complexes of Co(II), Ni(II), Cu(II) and Zn(II) were prepared in a two-step procedure. The first step included the mixing of an equal molar of metal sulfate hydrate and nitrilotriaceticacid in 70 mL of bi-distilled water and heated until the volume was greatly reduced. The binary metallic moieties in a 1:1 ratio were obtained by adding ethanol to the solution with scratching, followed by filtrating, washing with ethanol and drying in an oven at about 80 °C for 2 h. 

The ternary metal complexes in a 1:1:1 ratio were then prepared by mixing equivalent amounts of the prepared binary materials with glutamic acid in 80 mL of bi-distilled water. The same workup has been followed to get the desired complexes. The crystallization of the new complexes was achieved in bi-distilled water.

#### 3.3.2. Synthesis of Trivalent Metal Complexes of Cr or Fe

The trivalent metal complexes of Cr(III) or Fe(III) were carried out by combining equal molar ratios of basic chromium sulfate or ferric sulfate hydrate, (NTA) and glutamic acid in 100 mL of bi-distilled water and by heating until the volume is greatly reduced. Ethanol was added to acquire the ternary metal complexes with a 1:1:1 ratio. The product was isolated and crystalized, using the same procedure. 

#### 3.3.3. Synthesis of Nanosized Coordination Polymers

The nano-complexes were prepared by mixing 10 mL of each of the 10^−2^ M of metal sulfate hydrate or basic chromium sulfate, nitrilotriacetic acid and glutamic acid in a 50 mL beaker. The solution was heated, nearly boiled, until the volume was reduced to about 10 mL. Then, different concentrations of capping agents (10^−5^ or 10^−6^ M of PVA) or (10^−2^ or 10^−3^ M of CTAB) were added and left overnight. The obtained precipitate was filtered, washed with ethyl alcohol and dried under a vacuum.

### 3.4. In Silico Predictions

#### 3.4.1. Bioactivity Predictions Using Molinspiration

The complexes were assessed for predicting their in silico bioactivity by employing the online interface from the Molinspiration Chem informatics server (http://www.molinspiration.com). The Molinspiration render vast variety of chem informatics software tools involves the conversion of SMILES and SD file, the manipulation and processing of a molecule and the calculation of varied molecular properties vital for the QSAR studies, drug design, molecular modelling, etc. This server also braces virtual screening that is based on fragment, forecast of bioactivity and visualization of data. The tools for Molinspiration are written in Java; therefore, they can be practically entertained on any computer platform. The Molinspiration tool possesses an amiscreen engine, which first analyzes a training set of active structures and then makes a comparison with molecules that are inactive by utilizing the sophisticated Bayesian statistics.

Molinspiration calculates the bioactivity contribution of every single substructure fragment because this toll is fragment-based. In the designated window, the chemical structures are drawn directly, thus, predicting the bioactivity by calculating the sum of activity contributions made by the fragments of the targeted candidates. The molecular activity score was procured from it. Candidates, which are in the possession of the highest activity outcome, tend to have the highest likelihood to be active. 

On the basis of the described decorum above, screening models were developed to monitor various inhibitions, i.e., G protein-coupled receptors ligand (GPCR ligands), kinase, ion channel blockers/modulators, protease, nuclear receptor ligands and enzyme inhibition.

#### 3.4.2. Physicochemical and Pharmacokinetic Predictions Using Swiss ADME

The pharmacokinetic and physicochemical predictions for the complexes were executed by an online Swiss ADME tool of the Swiss Institute of Bioinformatics (http://www.sib.swiss). The input section itself is embraced with a molecular sketcher on the basis of Chem Axon’s Marvin JS, enabling the user to draw, import and edit a 2-D chemical structure. The chemical structures were inputted directly into the designated window and then the ADME, drug-likeness, physicochemical and pharmacokinetics properties were analyzed.

### 3.5. Biological Tests

#### 3.5.1. Cell Culture

The tumor cell lines breast adenocarcinoma (MCF-7), human colon carcinoma (HCT-116) and hepatocellular carcinoma (HepG-2) were derived from the American Type Culture Collection (ATCC, Rockville, MD). The cells were grown on the RPMI-1640 medium and supplemented with 10% of inactivated fetal calf serum and 50 µg/mL of gentamycin. The cells were maintained at 37 °C in a humidified atmosphere with 5% of CO_2_ and were subcultured two to three times a week. 

#### 3.5.2. Cytotoxicity Evaluation Using Viability Assay

The tumor cell lines were suspended in a medium at concentration 5 × 104 cell/well in Corning® 96-well tissue culture plates and then incubated for 24 h. The tested compounds with concentrations, ranging from 0 to 50 μg/mL, were then added into 96-well plates (six replicates) to achieve different concentrations for each compound. Six vehicle controls with a media of 0.5% DMSO were run for each of 96-well plates as a control. After incubating for 24 h, the number of viable cells were determined by the 3-(4,5-Dimethylthiazol-2-Yl)-2,5-Diphenyltetrazolium Bromide (MTT) test. Briefly, the media was removed from the 96-well plates and replaced with 100 μL of the fresh culture RPMI 1640 medium without phenol red, and 10 µL of the 12 mM MTT stock solution (5 mg of MTT in 1 mL of phosphate-buffered saline (PBS)) was added to each well, including the untreated controls. The 96-well plates were incubated at 37 °C with 5% of CO_2_ for 4 h. An 85-μL aliquot of the media was removed from the wells, and 50 µL of DMSO was added to each well, mixed thoroughly with the pipette and incubated at 37 °C for 10 min. Then, the optical density was measured at 590 nm with the microplate reader (Sunrise, TECAN, Inc, USA) to determine the number of viable cells, and the percentage of viability was calculated as (1 − (ODt/ODc)) × 100%, where the ODt is the mean optical density of wells treated with the tested sample and the ODc is the mean optical density of the untreated cells. The relation between the surviving cells and drug concentrations were plotted to get the survival curve of each tumor cell line after treatment with the specified compound. The 50% inhibitory concentration (IC_50_), the concentration required to cause toxic effects in 50% of intact cells, was estimated from graphic plots of the dose response curve for each concentration [65].

### 3.6. Antimicrobial Assay

The antimicrobial activities of the prepared complexes were screened for their in vitro antimicrobial activity at 25 µg/mL to determine the zone of inhibition against two gram-positive bacteria (Streptococcus pneumoniae and Bacillissubtilis), two gram negative bacteria (Pseudomonas aeruginosa and Escherichia coli) and four Fungi (Aspergillus fumigates, Syncephalastrumracemosum, Geotricumcandidum, Candida albicans). The activities of these compounds were tested by the agar diffusion method, using the Mueller–Hinton agar medium for bacteria and Sabouraud’s agar medium for fungi [64]. The tested compounds were dissolved in N,N-dimethylformamide (DMF) to give a solution of 1 mg ml^−1^. The inhibition zones (diameter of the hole) were measured in millimeters (6 mm) at the end of the incubation period of 48 h at 28 °C; N,N-dimethylformamide showed no inhibition zone.

## 4. Conclusions

A novel series of coordinated oligomers, incorporating octahedral ternary metal complexes, has been prepared via the reactions of M(II and III) sulfate with the NTA and glutamic acid. The sulfate ions acted as bridged bidentate ligands between the metal centers and granted the isolation of the oligomeric materials. The new materials have been characterized, using various techniques such as elemental analysis, FTIR, atomic absorption, mass spectrometry, thermal gravimetric analysis (TGA), UV-vis spectrometry and magnetic measurements. The in silico predictions revealed that the targeted units have good bioactivity against 4 of the 6 targeted enzymes and receptors. They have no BBB permeation but have low GI and skin permeation, which allows them to be considered as substrates for P-gp. The in vitro assay of the desired molecules revealed strong antimicrobial activity for some of the complexes, which assorted them as potential candidates for this purpose. In particular, copper and iron containing oligomers displayed stronger activity than the reference controls against three types of fungi (Aspergillusfumigatus, Syncephalastrumracemosum and Aspergillusfumigatus), against gram-positive Streptococcus pneumonia and Bacillissubtilis as well as against gram-negative Pseudomonas aeruginosa and Escherichia coli. In contrast, the antitumor inhibitory activity of these materials was not strong in general.

The same precursors have been utilized to isolate the nanosized analogues of the desired complexes through the use of CTAB and PVA as capping agents. The size of these complexes has been determined using the transmission electron microscope (TEM) that exhibited a highly mono-distribution behavior with an average size in the range of 2–9 nm under optimizing conditions. The scanning electron microscope (SEM) images of some of these complexes were also obtained and showed successful dispersion of the nano-molecules as discrete units. 

## Figures and Tables

**Figure 1 ijms-20-00743-f001:**
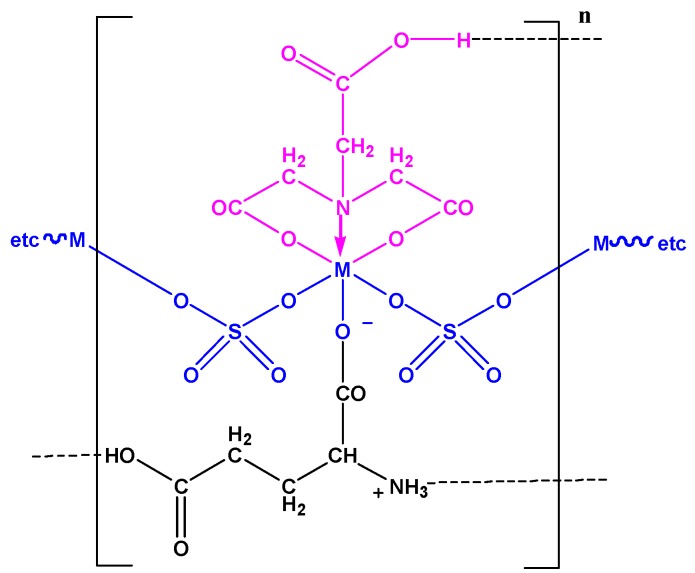
The expected octahedral structure and probable H-bond sites of the ternary metal oligomers: (M(HNTA)(GluH)(SO_4_)) complexes where *n* = −2 for M = Cu, Ni, Co and Zn and *n* = −1 for M = Cr and Fe.

**Figure 2 ijms-20-00743-f002:**
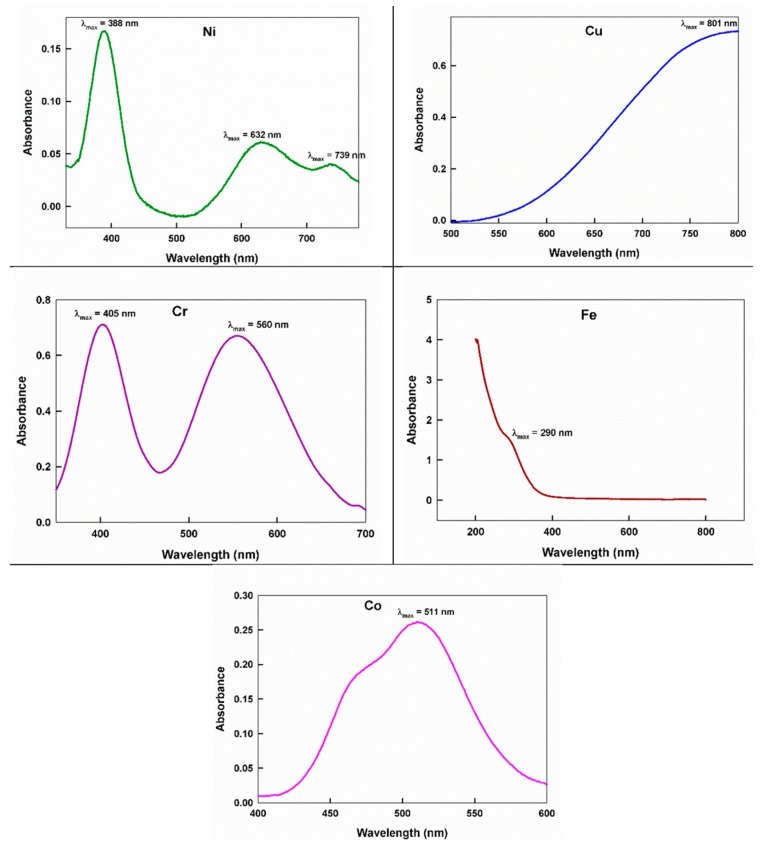
The UV-Vis spectrum of some of the coordination oligomers.

**Figure 3 ijms-20-00743-f003:**
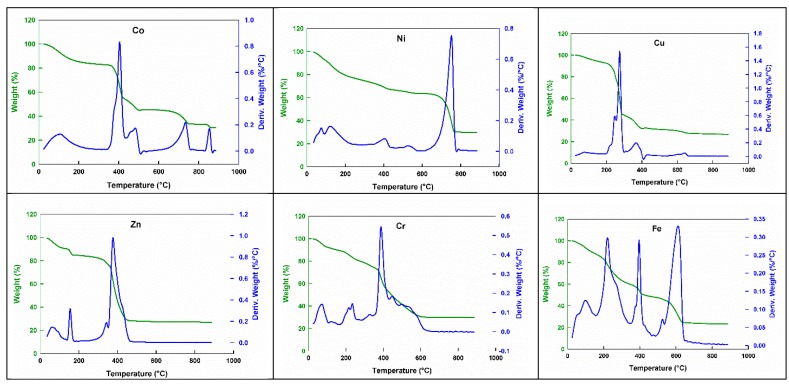
The thermograms of the ternary metal coordinated oligomers.

**Figure 4 ijms-20-00743-f004:**
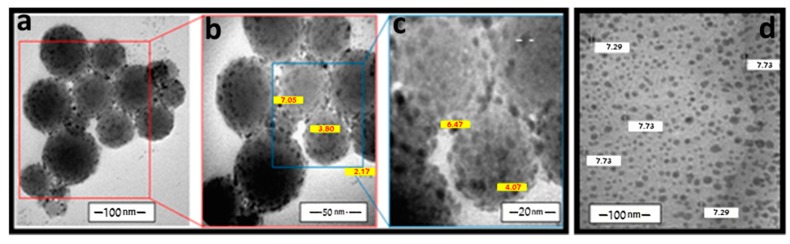
TEM images of nano nickel complex (**a**–**c**) and copper complex (**d**) in the presence of CTAB (10^−2^ M).

**Figure 5 ijms-20-00743-f005:**
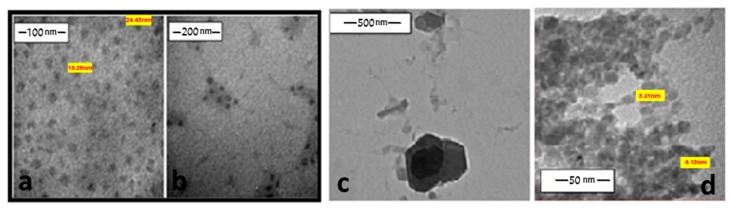
TEM images of nano copper complex with 10^−3^ M CTAB (**a**,**b**), nano nickel complex with 10^−2^ MCTAB (**c**) and nano zincwith 10^−2^ MCTAB (**d**).

**Figure 6 ijms-20-00743-f006:**
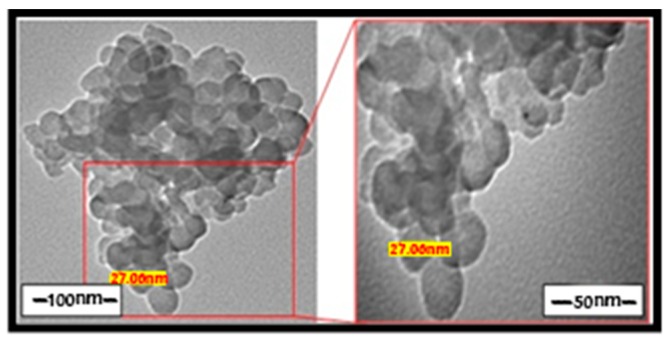
TEM images of some nano chromium complex in the presence of PVA (10^−5^ M).

**Figure 7 ijms-20-00743-f007:**
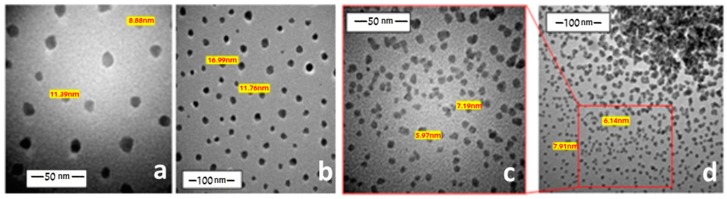
TEM images of some nano copper complex (**a**,**b**) and nano zinc (**c**,**d**) in the presence of PVA (10^−6^ M).

**Figure 8 ijms-20-00743-f008:**
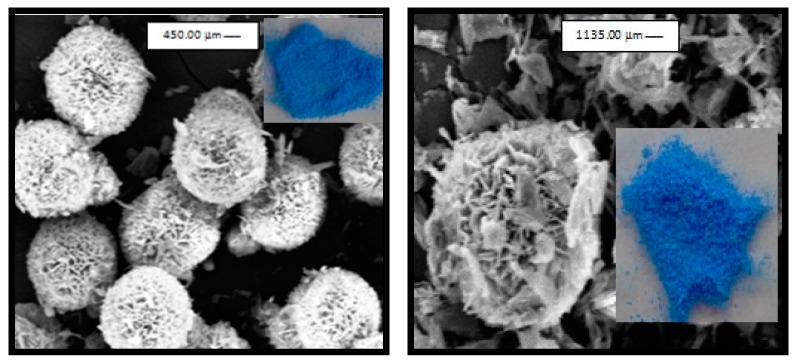
SEM images of nano-oligomer-containing copper complex with Glu in the presence of CTAB (10^−2^ M) (**left**) or PVA (10^−5^ M) (**right**).

**Table 1 ijms-20-00743-t001:** Some physico-properties of the prepared coordination oligomers.

No.	Monomeric Unit	Color	Mol. Wt.	pH	Magnetic Moment	C% cal/found	H% cal./found	N% cal./found	S% cal./found	M% cal./found
1	H_2_(Co(HNTA)(GluH)(SO_4_))	Pink	493.268	3.3	6.49	26.76/25.98	3.65/3.57	5.68/4.99	6.49/7.50	11.95/12.00
2	H_2_(Ni(HNTA)(GluH)(SO_4_))	Green	493.028	4	2.99	26.77/25.98	3.65/3.85	5.68/4.86	6.49/7.55	11.90/11.75
3	H_2_(Cu(HNTA)(GluH)(SO_4_))	Blue	497.881	4.4	1.16	26.51/26.96	3.62/2.95	5.62/5.38	6.43/5.74	12.76/12.87
4	H_2_(Zn(HNTA)(GluH)(SO_4_))	White	499.744	2.3	diamagnetic	26.41/26.50	3.60/3.32	5.60/5.31	6.40/6.70	14.09/14.56
5	H(Cr(HNTA)(GluH)(SO_4_))	Blue violet	486.331	3	4.96	27.14/27.49	3.70/3.70	5.75/5.26	6.58/8.21	10.69/10.49
6	H(Fe(HNTA)(GluH)(SO_4_))	Paige	490.182	3	4.16	26.93/26.86	3.67/3.65	5.71/5.40	6.53/7.50	13.39/13.62

**Table 2 ijms-20-00743-t002:** The infrared data of the prepared coordination oligomers.

No.	Ligands and Oligomers	υ (OH)	υ (NH_3_^+^)	υ (COOH)	υ (COO^−^)	υ (C–N)	υ (M–N)	υ (M–O)	ν_1_ (SO_4_)	ν_2_ (SO_4_)	ν_3_ (SO_4_)	ν_4_ (SO_4_)
	NTA	3434	-	1725	1575 (as) 1530 (s)	1206	-	-	-	-	-	-
	Glu	3434	3062	1728	1641 (as) 1418 (s)	1242	-	-	-	-	-	-
1	(H_2_(Co(HNTA)(GluH)(SO_4_)))n	3417	3048	-	1587 (as) 1455–1421 (s)	1026	551	395	989	446	1026 1177 1102	551 633
2	(H_2_(Ni(HNTA)(GluH)(SO_4_)))n	3395	3056	-	1589 (as) 1462–1414 (s)	1088	556	343	981	443	1064–1088 1185 1122	564 626 669
3	H_2_(Cu(HNTA)(GluH)(SO_4_)))n	3424	3193	-	1597–1567 (as) 1435–1418 (s)	1048	552	332	983	467	1048 1193 1102	596 604 641
4	(H_2_(Zn(HNTA)(GluH)(SO_4_)))n	3452	3048	1724	1622–1584 (as) 1461–1406 (s)	1064	557	318	988	477	1064 1169 1153	570 627 677
5	(H(Cr(HNTA)(GluH)(SO_4_)))n	3450	3056	-	1644 (as) 1435–1387 (s)	1051	550	330	995	479	1051 1185 1104	598 620 653
6	(H(Fe(HNTA)(GluH)(SO_4_)))n	3445	3080	-	1628–1548 (as) 1445 (s)	1044	556	320	997	482	1072 1169 1104	588 629 661

**Table 3 ijms-20-00743-t003:** Mass spectra data of the prepared coordination oligomers.

No.	Oligomers	Monomer Mol. Wt.	Monomer *m/z*	*m/z*
1	(H_2_(Co(HNTA)(GluH)(SO_4_)))n	493.268	497	1678
2	(H_2_(Ni(HNTA)(GluH)(SO_4_)))n	493.028	499	1321
3	(H_2_(Cu(HNTA)(GluH)(SO_4_)))n	497.881	495	1217
4	(H_2_(Zn(HNTA)(GluH)(SO_4_)))n	499.744	509	1307
5	(H(Cr(HNTA)(GluH)(SO_4_)))n	486.331	499	1627
6	(H(Fe(HNTA)(GluH)(SO_4_)))n	490.182	425	2402

**Table 4 ijms-20-00743-t004:** The values for the decomposition along with the species lost in each step.

No.	Oligomers	Mol. Wt.	TGA Range (°C)	Mass Loss (%)		Total Mass Loss (%)		Assignment
				Fou.	Calc.	Fou.	Calc.	
1	(H_2_(Co(HNTA) (GluH)(SO_4_)))n	493.268	25–150	10.50	10.34	74.20	75.41	H_2_S+NH_3_
			150–441	37.50	38.72			Maleic acid+glycine
			441–779	23.00	23.11			2CO_2_+C_2_H_2_
			779–868	3.20	3.24			CH_4_
			Above 868	25.80	24.92	Residue		CoSO_2_
2	(H_2_(Ni(HNTA) (GluH)(SO_4_)))n	493.028	25–127	10.00	10.34	70.50	72.20	H_2_S+NH_3_
			127–429	22.00	23.53			Maleic acid
			429–779	38.50	37.53			Glycine+3CO+C_2_H_2_
			Above 779	29.50	28.13	Residue		NiSO_3_
3	(H_2_(Cu(HNTA) (GluH)(SO_4_)))n	497.881	25–200	7.00	6.8	73.50	74.69	H_2_S
			200–664	66.50	67.89			Maleic acid+ glycine+ CH_4_+2CO_2_+C_2_H_2_ +NH_3_
			Above 664	26.50	25.62	Residue		CuSO_2_
4	(H_2_(Zn(HNTA) (GluH)(SO_4_)))n	499.744	25–135	10.00	10.21	73.00	74.44	H_2_S + NH_3_
			135–500	63.00	64.23			Maleic acid + glyciene + 2CO_2_ + C_2_H_2_+ CH_4_
			Above 500	26.90	25.89	Residue		ZnSO_4_
5	(H(Cr(HNTA) (GluH)(SO_4_)))n	486.331	25–100	7.00	6.99	70.50	71.26	H_2_S
			100–212	6.50	6.79			NH_3_ +CH_4_
			212–650	57.00	56.55			Maleic acid + Glycine + 3CO
			Above 650	29.50	30.43	Residue		CrSO_4_
6	H(Fe(HNTA) (GluH)(SO_4_))	490.182	25–122	7.00	6.94	77.00	75.9	H_2_S
			122–400	41.00	39.99			Glycine + 2CO_2_ + NH_3_ + CH_4_
			400–650	29.00	28.97			Maleic acid + C_2_H_2_
			Above 650	23.00	24.45	Residue		FeSO_4_

**Table 5 ijms-20-00743-t005:** The bioactivity score profiles of the selected compounds (1–6)**.**

No.	GPCR Ligand	Ion Channel Modulator	Kinase Inhibitor	Nuclear Receptor Ligand	Protease Inhibitor	Enzyme Inhibitor
1	0.04	0.21	−0.00	0.01	0.47	0.50
2	0.04	0.21	−0.00	0.01	0.47	0.50
3	0.04	0.21	−0.00	0.01	0.47	0.50
4	0.40	0.21	−0.00	0.01	0.47	0.54
5	0.04	0.21	−0.00	0.01	0.47	0.50
6	0.45	0.21	−0.00	0.01	0.47	0.53

**Table 6 ijms-20-00743-t006:** Physicochemical properties of the selected compounds (1–6).

No.	Monomeric Unit	Mol. wt.	Fraction Csp3 ^a^	HBA ^b^	HBD ^c^	Molar Refractivity	Water Solubility	TPSA ^d^
1	H_2_(Co(HNTA)(GluH)(SO_4_))	493.268	0.58	19	7	100.09	Yes	364.14
2	H_2_(Ni(HNTA)(GluH)(SO_4_))	493.028	0.57	19	7	100.15	Yes	364.16
3	H_2_(Cu(HNTA)(GluH)(SO_4_))	497.881	0.55	19	7	100.15	Yes	364.17
4	H_2_(Zn(HNTA)(GluH)(SO_4_))	499.744	0.56	19	7	100.11	Yes	364.13
5	H(Cr(HNTA)(GluH)(SO_4_))	486.331	0.54	19	7	100.13	Yes	364.15
6	H(Fe(HNTA)(GluH)(SO_4_))	490.182	0.55	19	7	100.15	Yes	364.17

^a^ The ratio of sp^3^ hybridized carbons over the total carbon count of the molecule, ^b^ number of hydrogen bond acceptors, ^c^ number of hydrogen bond donors and ^d^ topological polar surface area (Å^2^).

**Table 7 ijms-20-00743-t007:** Pharmacokinetic predictions of the selected compounds (1–6).

No.	Monomeric Unit	GI abs ^a^	BBB Permeant ^b^	P-Gpsubstrate ^c^	CYP1A2 Inhibitor ^d^	Log Kp ^e^
1	H_2_(Co(HNTA)(GluH)(SO_4_))	Low	No	Yes	No	−16.21
2	H_2_(Ni(HNTA)(GluH)(SO_4_))	Low	No	Yes	No	−16.21
3	H_2_(Cu(HNTA)(GluH)(SO_4_))	Low	No	Yes	No	−16.24
4	H_2_(Zn(HNTA)(GluH)(SO_4_))	Low	No	Yes	No	−16.25
5	H(Cr(HNTA)(GluH)(SO_4_))	Low	No	Yes	No	−16.17
6	H(Fe(HNTA)(GluH)(SO_4_))	Low	No	Yes	No	−16.19

^a^ Gastro Intestinal absorption, ^b^ Blood Brain Barrier permeant, ^c^ P-glycoprotein substrate, ^d^ CYP1A2: Cytochrome P450 family 1 subfamily A member 2 (PDB:2HI4) and ^e^ skin permeation in cm/s.

**Table 8 ijms-20-00743-t008:** The antimicrobial activity of the prepared ternary metal complexes containing oligomers.

No.	Cpd.	*S. pneum.* (+ve)	*B. Subtilis* (+ve)	P. *Aerug.* (−ve)	*E. coli* (−ve)	*A. fumig.*	*S. raceme.*	*G. candi.*	*C. albic.*
	Amphotericin B	NA	NA	NA	NA	23.7 ± 0.1 (0.24)	19.7 ± 0.2 (3.9)	28.7 ± 0.2 (0.015)	25.4 ± 0.1 (0.12)
	Ampicillin	23.8 ± 0.2 (0.24 )	32.4 ± 0.3 (0.007 )	NA	NA	NA	NA	NA	
	Gentamicin	NA	NA	17.3 ± 0.1 (15.63 )	19.9 ± 0.3 (3.9 )	NA	NA	NA	NA
1	(H_2_(Co(HNTA) (GluH)(SO_4_)))n	NA	12.3 ± 0.37	NA	13.6 ± 0.58	13.4 ± 0.63	11.6 ± 0.58	12.6 ± 0.25	10.6 ± 0.44
2	(H_2_(Ni(HNTA) (GluH)(SO_4_)))n	22.0 ± 0.29 (1.95)	26.1 ± 0.44 (0.24)	22.9 ± 0.58 (7.81)	21.9 ± 0.72 (1.95)	20.3 ± 0.58 (3.9)	19.4 ± 0.44(31.25)	22.3 ± 0.63 (3.9)	18.4 ± 0.44 (125)
3	(H_2_(Cu(HNTA) (GluH)(SO_4_)))n	20.9 ± 0.44 (1.95)	23.8 ± 0.17 (0.24)	22.3 ± 0.17 (0.98)	25.2 ± 0.63 (0.12)	24.8 ± 0.58 (0.12)	23.8 ± 0.58 (0.24)	24.2 ± 0.44 (0.24)	20.9 ± 0.37 (1.95)
4	(H_2_(Zn(HNTA) (GluH)(SO_4_)))n	18.2 ± 0.63 (15.63)	19.3 ± 0.72 (7.81)	NA	11.8 ± 0.58 (500)	18.4 ± 0.58 (15.63)	15.2 ± 0.58 (62.5)	15.9 ± 0.08 (62.5)	9.8 ± 0.44 (NA)
5	(H(Cr(HNTA) (GluH)(SO_4_)))n	14.5 ± 0.44 (62.5)	22.3 ± 0.37 (0.98)	NA	11.6 ± 0.25 (125)	17.2 ± 0.63 (31.25)	16.8 ± 0.44 (31.25)	18.3 ± 0.44 (15.63)	NA
6	(H(Fe(HNTA) (GluH)(SO_4_)))n	24.7 ± 0.58 (0.12)	28.2 ± 0.58 (0.015)	22.8 ± 0.44 (0.49)	24.4 ± 0.63 (0.24)	24.4 ± 0.37 (0.24)	17.2 ± 0.72 (31.25)	25.3 ± 0.44 (0.12)	19.9 ± 0.77 (3.9)

* NA is no activity, and the values in brackets are the MIC values.

**Table 9 ijms-20-00743-t009:** The cytotoxicity assays of the new oligomers against MCF-7, HCT-116 and HepG-2 cell lines.

No.	Oligomers	Sample Conc. (µg/mL)	MCF-7 Viability%	IC50 (µg/mL)	HCT-116 Viability (µg/mL)	IC50 (µg/mL)	HepG-2 Viability%	IC50 (µg/mL)
1	(H_2_(Co(HNTA) (GluH)(SO_4_)))n	50	64.28	>50	68.57	>50	60.86	>50
		25	72.73		82.46		78.13	
		12.5	89.42		93.12		91.48	
		6.25	94.54		98.78		97.24	
		3.125	98.16		100		100	
		1.56	100		100		100	
		0	100		100		100	
2	(H_2_(Ni(HNTA) (GluH)(SO_4_)))n	50	49.84	49.7 ± 0.23	57.42	>50	46.18	44.8 ± 0.24
		25	63.51		69.78		64.39	
		12.5	71.62		83.14		80.61	
		6.25	82.94		92.65		91.74	
		3.125	91.05		98.78		96.52	
		1.56	97.48		100		99.06	
		0	100		100		100	
3	(H_2_(Cu(HNTA) (GluH)(SO_4_)))n	50	24.32	11.5 ± 0.09	26.98	18.00 ± 0.08	21.54	12.00 ± 0.05
		25	33.57		39.44		36.93	
		12.5	45.34		58.23		48.22	
		6.25	74.69		75.16		71.36	
		3.125	85.71		89.28		84.59	
		1.56	93.56		95.34		92.64	
		0	100		100		100	
4	(H_2_(Zn(HNTA) (GluH)(SO_4_)))n	50	20.43	10.7 ± 0.11	29.17	20.1 ± 0.10	23.67	16.1 ± 0.06
		25	34.68		45.26		40.89	
		12.5	42.97		57.41		53.72	
		6.25	67.25		72.41		69.17	
		3.125	80.13		81.04		86.28	
		1.56	87.49		89.56		93.02	
		0	100		100		100	
5	(H(Cr(HNTA) (GluH)(SO_4_)))n	50	77.18	>50	84.62	>50	76.28	>50
		25	86.27		93.13		91.36	
		12.5	95.84		98.74		97.96	
		6.25	99.08		100		100	
		3.125	100		100		100	
		1.56	100		100		100	
		0	100		100		100	
6	(H(Fe(HNTA) (GluH)(SO_4_)))n	50	45.96	47.3 ± 0.19	61.84	>50	42.69	44.1 ± 0.17
		25	82.74		83.12		73.48	
		12.5	90.36		91.46		82.97	
		6.25	94.28		97.38		91.64	
		3.125	98.93		99.17		98.26	
		1.56	100		100		100	
		0	100		100		100	

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
