# Peer review of "Design of Novel Oligomeric Mixed Ligand Complexes: Preparation, Biological Applications and the First Example of Their Nanosized Scale"

_ijms, 2019, doi:10.3390/ijms20030743_

Reviewer 1 Report

In the manuscript “Design of Novel oligomeric mixed ligand complexes: preparation, biological applications and the first example of their nanosized scale”,  Okasha et al  prepared the coordinated oligomers together with octahedral ternary metal complex.  They found that the new materials possess bioactivity against multiple enzymes and receptors. It is an interesting manuscript. However, I have some concerns.  

How’s the      reproducibility of Figures 6 and 7.

Section      2.9: “This behavior indicates that a higher concentration of the CTAB is      considered more suitable for the preparation of these nano-complexes. “.      The authors should mentioned that the conclusion is based on the      concentration they studies. This conclusion may not work for the high      concentration of CTAB.

Introduction:      “ Coordination compounds and macromolecules are an eminent class of      material that are encountered daily. “ More references (e.g. e.g. The Journal      of Chemical Physics, 2009, 131, 244904; The Journal of Chemical Physics,      2018, 149, 174705; Polymer, 2010, 51, 5869-5882; Journal of Physical Chemistry C, 2018, 122,  2712-2716 ; Journal of Colloid and Interface Science,      2011, 363, 573-578; Langmuir, 2014, 30, 3723-3728) are needed to support      this statement.

Please      provide clear pictures for Figures 2 and 3.

Please      provide clearer labels in Figures 6, 7, 8, 9 and 10.

Author Response

We appreciate the comments raised by the reviewer and the following part is a clarification for the addressed points:

1.      How’s the reproducibility of Figures 6 and 7.   

The reviewer made a good point. The nanomaterials in Figures 6-9 (Figures 4-7 in the modified manuscript) are reproducible. The experiment have been repeated several times (five to seven times). In fact, we are submitting further data with different capping ligands for publication, which displays a successful preparation of well-dispersed nanoparticles.   

2.      Section 2.9: “This behavior indicates that a higher concentration of the CTAB is considered more suitable for the preparation of these nano-complexes. “. The authors should mentioned that the conclusion is based on the concentration they studies. This conclusion may not work for the high concentration of CTAB.

We followed the reviewer’s advice; the sentence has been modified in the text. (line 306, Revised Manuscript). 

This behavior indicates that a higher concentration of the CTAB, within the studied concentrations, is considered more suitable for the preparation of these nano-complexes.

3.      Introduction: “Coordination compounds and macromolecules are an eminent class of material that are encountered daily”. More references are needed to support this statement.

We thank the reviewer for this comment; the required references have been added. (Ref. No. 2-7, Revised Manuscript).

4.      Please provide clear pictures for Figures 2 and 3.

We appreciate the reviewer comments; Figures 2 and 3 have been modified.

5.      Please provide clearer labels in Figures 6, 7, 8, 9 and 10.

We followed the reviewer’s comment; the labels on each of the figures have been clarified, Figures 4-8 in the modified manuscript.

Reviewer 2 Report

The paper is devoted to the synthesis of novel coordinated oligomers, incorporating octahedral ternary metal complexes, via the reactions of metal (II and III) sulfates with the NTA and glutamic acid. The new materials have been characterized using elemental analysis, FTIR, atomic absorption, mass spectrometry, thermal gravimetric analysis (TGA), UV-vis spectrometry, and magnetic measurements; the biological activity has been studied as well.

The authors have done quite a lot of work. The article is clearly presented, and it is made on a good experimental and theoretical level. Nevertheless, it needs to be improved.

1) The Introduction contains 57 (!) references, while the total number of Refs. in the article is 75. The citations in the Introduction should be reduced.

2) In the Abstract it is necessary to specify the metals, which were used as a base for the oligomer synthesis

3) Line 120: what does it mean: n = -2 for M = Cu, Ni, Co, Zn and n= -1 for M = Cr , Fe?

4) Line 141. The number of monomeric units in oligomers is the most serious problem of this article. Here, the mass spectra should be given, and in the experimental part it should described in more detail how the mass spectra were obtained, because the direct input was probably used. The given masses > 1000 correspond to fractional values of n. Why is this so? Methodically it would be more correct to obtain the MALDI spectra.

5) Line 150: misprint – “Mmonomer”

6) Line 215: Is this about predicted bioactivity?

7) Figures 4 and 5 should be removed, because the information is repeated in Table 8.

8) For IC50 the errors must be evaluated.

9) The quality of figures 6-8 is poor. Inscriptions highlighted in yellow are practically unreadable.

Author Response

We appreciate the comments raised by the reviewer and the following part is a clarification for the addressed points:

1.      The citations in the Introduction should be reduced.

We have considered the comments of reviewer. The citations have been reduced. 

2.      In the Abstract it is necessary to specify the metals, which were used as a base for the oligomer synthesis

We appreciated the reviewer remark; the used metals have been specified in the abstract.

3.      Line 120: what does it mean: n = -2 for M = Cu, Ni, Co, Zn and n= -1 for M = Cr , Fe

We thank the reviewer for the comment; n = -2 for M = Cu, Ni, Co, and Zn and n= -1 for M = Cr, and Fe are the charge on the repeating units of the complexes, and it has been corrected on the figure.

4.      Line 141. The number of monomeric units in oligomers is the most serious problem of this article. Here, the mass spectra should be given, and in the experimental part it should described in more detail how the mass spectra were obtained, because the direct input was probably used. The given masses > 1000 correspond to fractional values of n. Why is this so? Methodically it would be more correct to obtain the MALDI spectra.

We appreciate the reviewer’s remarks. The mass spectra have been measured, using two different mass spectrophotometers. The first characterization was recorded at 350oC and 70 eV on the Shimadzu GC/MS-QP5050A spectrometer and exhibited the ionic peaks of the monomeric units. Further mass spectra analysis has been performed on these molecules, using a triple quadruple mass spectrometry API4500 (ABSciex, Canada), attached to the Exion LC™ chromatographic system (ABSciex, USA), in order to prove the fragmentation of the oligomeric units, which can be seen in the attached file. In actuality, the MALDI instrumentation is not available, so we couldn’t have this analysis; however, thus far, our data has revealed that we have multiple repeating units (oligomerization) of these complexes. The experimental has been clarified (line 352-359, modified manuscript).

5.      Line 150: misprint – “Mmonomer”.

We thank the reviewer for the advice, and this word has been modified

6.      Line 215: Is this about predicted bioactivity?

We thank the reviewer for the comment; line 215 is caption of the thermogravimetric analysis of the prepared oligomers.  

7.      Figures 4 and 5 should be removed, because the information is repeated in Table 8.

We appreciate the reviewer for the comment. The figures have been omitted.

8.      For IC50 the errors must be evaluated.

We thank the reviewer for their comments. The values of the errors have been added.

9.      The quality of figures 6-8 is poor. Inscriptions highlighted in yellow are practically unreadable.We appreciated the reviewer comments; the figures and labels have modified and clarified, Figures 4-7 in the modified manuscript.

Round  2

Reviewer 1 Report

The authors have considered all points I raised in my initial report. The manuscript is improved. I recommend the publication of this manuscript

Reviewer 2 Report

The authors did a good job of improving the quality of the article.

One minor remark remains.

In the old version - line 215 and in the new version line 229: The caption to the Table 5 should be changed on "Predicted bioactivity...."